# Long-Term Impairment of Working Ability in Subjects under 60 Years of Age Hospitalised for COVID-19 at 2 Years of Follow-Up: A Cross-Sectional Study

**DOI:** 10.3390/v16050688

**Published:** 2024-04-26

**Authors:** Luisa Frallonardo, Annunziata Ilenia Ritacco, Angela Amendolara, Domenica Cassano, Giorgia Manco Cesari, Alessia Lugli, Mariangela Cormio, Michele De Filippis, Greta Romita, Giacomo Guido, Luigi Piccolomo, Vincenzo Giliberti, Francesco Cavallin, Francesco Vladimiro Segala, Francesco Di Gennaro, Annalisa Saracino

**Affiliations:** 1Department of Precision and Regenerative Medicine and Ionian Area (DiMePRe-J), Clinic of Infectious Diseases, University of Bari “Aldo Moro”, Piazza Giulio Cesare n. 11 Cap, 70124 Bari, Italy; a.ileniaritacco@gmail.com (A.I.R.); angelaamendolara9@gmail.com (A.A.); menicacassano@gmail.com (D.C.); mancocesarigiorgia@gmail.com (G.M.C.); alessialugli@hotmail.it (A.L.); mariangela.cormio@hotmail.it (M.C.); m.defilipippis@studenti.uniba.it (M.D.F.); greta.romit@gmail.com (G.R.); giacguido@gmail.com (G.G.); lpiccolomo@outlook.it (L.P.); vincenzogiliberti8@gmail.com (V.G.); fvsegala@gmail.com (F.V.S.); francesco.digennaro1@uniba.it (F.D.G.); annalisa.saracino@uniba.it (A.S.); 2Independent Statistician, 36020 Solagna, Italy; cescocava@libero.it

**Keywords:** COVID-19, chronic fatigue syndrome, PASC, Post-Acute Sequelae of SARS-CoV-2 infection, post-COVID syndrome, SARS-CoV-2

## Abstract

**Background:** Coronavirus disease 2019 (COVID-19) can lead to persistent and debilitating symptoms referred to as Post-Acute sequelae of SARS-CoV-2 infection (PASC) This broad symptomatology lasts for months after the acute infection and impacts physical and mental health and everyday functioning. In the present study, we aimed to evaluate the prevalence and predictors of long-term impairment of working ability in non-elderly people hospitalised for COVID-19. **Methods:** This cross-sectional study involved 322 subjects hospitalised for COVID-19 from 1 March 2020 to 31 December 2022 in the University Hospital of Bari, Apulia, Italy, enrolled at the time of their hospital discharge and followed-up at a median of 731 days since hospitalization (IQR 466–884). Subjects reporting comparable working ability and those reporting impaired working ability were compared using the Mann-Whitney test (continuous data) and Fisher’s test or Chi-Square test (categorical data). Multivariable analysis of impaired working ability was performed using a logistic regression model. **Results:** Among the 322 subjects who were interviewed, 184 reported comparable working ability (57.1%) and 134 reported impaired working ability (41.6%) compared to the pre-COVID-19 period. Multivariable analysis identified age at hospital admission (OR 1.02, 95% CI 0.99 to 1.04), female sex (OR 1.90, 95% CI 1.18 to 3.08), diabetes (OR 3.73, 95% CI 1.57 to 9.65), receiving oxygen during hospital stay (OR 1.76, 95% CI 1.01 to 3.06), and severe disease (OR 0.51, 95% CI 0.26 to 1.01) as independent predictors of long-term impaired working ability after being hospitalised for COVID-19. **Conclusions:** Our findings suggest that PASC promotes conditions that could result in decreased working ability and unemployment. These results highlight the significant impact of this syndrome on public health and the global economy, and the need to develop clinical pathways and guidelines for long-term care with specific focus on working impairment.

## 1. Introduction

PASC (Post-Acute Sequelae of COVID-19) is a heterogeneous clinical syndrome that persists or develops after acute SARS-CoV-2 infection and affects multiple organ systems [1].

According to the Centers for Disease Control and Prevention (CDC), health issues related to PASC can impact subjects who experienced mild, moderate, or severe episodes of COVID-19, as well as those who had asymptomatic cases, can last anywhere from weeks to years, and complications may lead to a disability [1]. The various symptoms related to PASC have been grouped into clusters (neurological, psychiatric, constitutional, gastrointestinal, musculoskeletal), and among these, some have a significant impact on the patient’s ability to work [2]. According to the last International Labor Organisation (ILO) report [3], the economic and labor crisis caused by COVID-19 could increase worldwide unemployment by almost 25 million. The report numbers are not homogeneous among low- and high-income countries. Nevertheless, the estimates suggest an increase in global unemployment ranging from 5.3 to 24.7 million. Given the potential for severe symptoms that can negatively affect both physical and mental well-being [4,5,6], including cognitive and neuropsychological impairments [7], as well as work performance in various professional settings, certain authors have sought to examine the influence of COVID-19 sequelae on employment status [8].This feature is critical, particularly when considering the long-term effects of COVID-19 on the younger and working population, which will have significant economic and social consequences. Few data are available on work impairment as a consequence of the SARS-CoV-2 infection; for this reason, we conducted a cross-sectional study to investigate the long-term impairment of working ability in individuals under the age of 60 who were hospitalized for COVID-19.

## 2. Materials and Methods

### 2.1. Study Design

This was a cross-sectional study on long-term impairment of working ability in non-elderly people hospitalized for COVID-19. Phone interviews were carried out from 14 February to 22 March 2023.

A signed consent for all eligible subjects was acquired during hospitalization (retrospective data) while verbal consent was registered in the telephone interview (prospective data). The study was approved by the Local Ethical Committee (number 7280, 04/2022).

### 2.2. Subjects

All subjects aged 60 or younger who were hospitalized for COVID-19 from 1 March 2020 to 31 December 2022 in a COVID-19 designed hospital (University Hospital of Bari, Apulia, Italy) were eligible for inclusion in the study. Subjects with mental disorders, those with chronic diseases that interfered with working ability, and those who were unemployed at the time of the study were excluded.

### 2.3. Procedures

We investigated signs/symptoms of PASC and working ability according to current literature on the topic [9,10,11]. All questions were posed as yes/no by phone interview from 14 February to 22 March 2023.

### 2.4. Endpoints

The objective of the study is to evaluate the prevalence and predictors of long-term impairment of working ability in non-elderly individuals hospitalized for COVID-19. It is essential to articulate the specific research questions or hypotheses guiding our investigation. Our study aims to explore:

The prevalence of long-term impairment of working ability among non-elderly individuals hospitalized for COVID-19:Impaired working ability compared to pre-COVID-19 period as reported by the participants.Predictors or risk factors associated with long-term impairment of working ability in this population, Demographic, clinical, or socio-economic factors that significantly influence the likelihood of long-term working impairment following COVID-19 hospitalization.The correlation between comorbidities, the severity of COVID-19 illness during hospitalization, and long-term working ability outcomes.

### 2.5. Statistical Analysis

Continuous data were summarized as median and interquartile range (IQR). Data were compared among groups using the Mann-Whitney test (continuous data) and Fisher’s test or Chi-Square test (categorical data). Multivariable analysis of impaired working ability was performed using a logistic regression model. The initial model included a set of clinically relevant candidate predictors (age at hospital admission, sex, hypertension, diabetes, obesity, any immunosuppression condition, COVID-19 vaccination status at admission, remdesivir, oxygen during hospital stay, severe disease and length of hospital stay) and the final model was selected using the Akaike Information Criterion (AIC) reduction procedure. The effect sizes were reported as odds ratio (OR) with 95% confidence interval (CI). All tests were two-sided, and a *p*-value < 0.05 was considered statistically significant.

Statistical analysis was carried out using R 4.3 (R Foundation for Statistical Computing, Vienna, Austria) [12].

### 2.6. Questionnaires

We administered the Hospital Anxiety and Depression Scale (HADS) surveys on Post-Traumatic Stress Disorder (PTSD).

The Hospital Anxiety and Depression Scale (HADS) is a widely used self-assessment tool designed to measure levels of anxiety and depression. It consists of 14 items, 7 for anxiety (HADS-A) and 7 for depression (HADS-D). The symptoms investigated in HADS-A include feeling tense, being concerned about several matters, feeling restless, having sudden feelings of panic, and feeling scared or nervous. The symptoms investigated in HADS-D include feeling unhappy or depressed, experiencing diminished capacity for pleasure, losing interest in appearance, and experiencing a sense of culpability for all occurrences.

Surveys on Post-Traumatic Stress Disorder (PTSD) cover several critical areas related to both conditions: traumatic events, PTSD symptoms such as intrusive memories, avoidance behaviors, adverse changes in thinking and mood, changes in reactivity and arousal, and sleep patterns and disorder and their impact on daily functioning.

## 3. Results

We evaluated for inclusion all 510 subjects aged 18–60 years who were hospitalized for COVID-19 from 1 March 2020 to 31 December 2022. After excluding 10 dead subjects and 6 subjects unfit for the study (having mental disorders or chronic diseases interfering with working ability), we attempted to contact the remaining 494 subjects, and we were able to interview 322 of them (response rate 65.2%). The interview was undertaken at a median of 731 days since hospitalization (IQR 466–884). The comparison of reachable and unreachable subjects is shown in Table 1. The difference in age between and the difference in gender distribution between the two groups are not statistically significant (*p* = 0.42), (*p* = 0.17). The prevalence of hypertension, dyslipidemia, diabetes, and obesity (BMI > 30 kg/m^2^) is not significantly different between reachable and unreachable subjects (*p* > 0.05). There is a significant difference in smoking habits between reachable and unreachable subjects (*p* = 0.01). Current smoking habits are more prevalent among unreachable subjects (11.1%) compared to reachable subjects (5.6%).

There is a significant difference in the use of high-flow nasal cannula during hospital stay, with a higher prevalence among reachable subjects (13.3%) compared to unreachable subjects (7.0%) (*p* = 0.04). The prevalence of ICU admission, oxygen use during hospital stay, non-invasive mechanical ventilation, and invasive mechanical ventilation is not significantly different between reachable and unreachable subjects (*p* > 0.05). The length of hospital stay is significantly longer for reachable subjects (median: 25 days, range: 19–40 days) compared to unreachable subjects (median: 11 days, range: 7–17 days) (*p* < 0.0001) (Table 1).

Among the 322 subjects who were interviewed, 184 reported comparable working ability (57.1%), and 134 reported impaired working ability (41.6%) compared to the pre-COVID-19 period, while four subjects were unemployed and unable to provide such information (1.2%). Individuals with impaired working ability had a higher median age at hospital admission (50 years, range: 44–54) compared to those with comparable working ability (44 years, range: 33–53). The difference in age between the two groups was statistically significant (*p* = 0.001).

The interview was undertaken at a median of 725 days since hospitalization (IQR 526–869) in subjects with comparable working ability and 749 days (IQR 432–1008) in those with impaired working ability (*p* = 0.61). The percentage of males was higher among individuals with comparable working ability (64.1%) compared to those with impaired working ability (51.5%). This difference was statistically significant (*p* = 0.03). Older age (*p* = 0.001), pre-existing conditions such as hypertension (*p* = 0.01), diabetes (*p* = 0.002), and smoking habits were more prevalent among individuals with impaired working ability compared to those with comparable working ability, and these differences were statistically significant (*p* < 0.05) (Table 2).

Receiving oxygen during hospital stay (*p* = 0.04) and longer hospital stay (*p* = 0.04) were more frequent among subjects with impaired working ability (Table 2).

Multivariable analysis identified age at hospital admission (OR 1.02, 95% CI 0.99 to 1.04), female sex (OR 1.90, 95% CI 1.18 to 3.08), diabetes (OR 3.73, 95% CI 1.57 to 9.65), receiving oxygen during hospital stay (OR 1.76, 95% CI 1.01 to 3.06) and severe disease (OR 0.51, 95% CI 0.26 to 1.01) as independent predictors of long-term impaired working ability after being hospitalized for COVID-19 (Table 3).

At the interview, median self-perception of overall health status was 7 (IQR 6–8) in subjects with impaired working ability and 8 (IQR 7–9) in those with comparable working ability (*p* < 0.0001). In addition, subjects with impaired working ability more frequently reported issues concerning psychiatric, neurological, respiratory, constitutional, skeletal muscle, gastrointestinal, or other symptoms (all *p* < 0.001), as well as other symptoms no longer present at the time of the interview (*p* = 0.0002) (Figure 1). Full results are reported in Table 4.

## 4. Discussion

PASC syndrome represents an important public and global health issue with a relevant impact on mortality and quality of life in people after SARS CoV-2 infection [13]. Despite much evidence of pathophysiology, epidemiology, and risk factors of PACS [14], few studies have shown the potential impact of PACS on work impairment in the working-age population with possible economic and social consequences. Nonetheless, the impact of chronic sequelae on the workforce and consequently on families’ finances continues to be a cause for concern, particularly in low-middle-income countries [11] (Appendix A).

For this reason, we performed a cross-sectional study to assess the impact of PACS on work capability in people younger than 60 y hospitalised due to COVID-19.

In our study, a relevant finding was the high occurrence of persistent job impairment in 41.6% of patients, seen after a median period of 725 days following hospitalization for COVID-19. This suggests that the burden of PASC may affect both quality of life and the workforce, making individuals more vulnerable to unemployment in the two years following hospitalization for SARS-CoV-2 infection. From this standpoint, certain characteristics can serve as predictors of long-term diminished functioning capacity, such as advanced age upon admission, female gender, diabetes, and the utilization of oxygen therapy during the hospitalization period [15] (Table 3).

As shown in several studies, males have a significantly higher risk of severe disease in the acute phase of SARS-CoV-2 infection, while females may be more prone to increased risk of developing PASC [16,17,18]. In addition, some researchers have tried to explain this phenomenon and have correlated the role of female hormones with a more constant hyper-inflammation also after the negativization of SARS CoV-2. Although a stronger and earlier production of IgG antibodies would explain the lower risk of mortality among females, it might also play a role in perpetuating disease manifestations with consequent PACS.

Several studies have attempted to characterize possible predictors of PASC. This syndrome can be explained in terms of the balance between damage and repair mechanisms, as well as the patient’s health-related quality of life and well-being at a patient-centered approach level. Damage and repair mechanisms are components of frailty and resilience, while well-being is part of the multidimensional conception [19].

In our study, older age at hospital admission was associated with increased risk for impaired working ability. The literature does not offer homogeneous data concerning the role of age in developing PASC. However, Cheng Lai et al. reported an association between the development of PASC-related chronic fatigue and advanced age [20,21,22], and a recent meta-analysis found that individuals aged 40–69 years and ≥70 years were more vulnerable to PASC compared to adult patients under the age of 40 [23].

The association between diabetes as a pre-existing comorbidity and the increased susceptibility to SARS-CoV-2 infection has been clinically recognized since the beginning of the pandemic [24]. Several findings concerning alterations in the glycemic balance during acute infection, vascular damage and the resulting alterations, and the pancreatic insult triggered by the cytopathic action of the virus [25], established a bidirectional relationship between stress-induced hyperglycemia and COVID-19 [26]. In agreement with previous studies [27], we found that diabetes was associated with increased risk for impaired working ability. Of note, there is emerging evidence suggesting a new potential risk of developing new-onset diabetes in the post-COVID period. This may represent a cornerstone in the clinical approach to the patient with PASC syndrome, both in terms of clinical management, early diagnosis, and prevention of complications of diabetes [28].

In contrast to previous findings, showing a dominant prevalence of respiratory symptoms in patients who received ventilatory support in hospital, interestingly, in our results, experiencing a “severe disease”, defined as receiving a high-flow nasal cannula or mechanical ventilation during hospitalization, actually has a protective effect. In line with other studies, our finding indicates that administering high-flow oxygen treatment during the acute phase of COVID-19, preventing oxygen deficiency in various tissues, may have a protective effect against post-infection sequelae [29]. This association can be related to the increased frequency of post-discharge health evaluations by suggesting that many mechanisms responsible for the post-COVID-19 state are not directly associated with acute lung injury but depend on multi-factorial processes [30].

Several pathological observations in individuals with post-COVID-19 condition show characteristics or overlap with the symptoms experienced by individuals with Chronic Fatigue Syndrome [11]. These observations involve alterations in the immunological, cardiovascular, metabolic, gastrointestinal, neurological, and autonomic systems [31]. In our study, individuals with long-term impaired working ability reported more frequently a series of symptoms that may affect their work performance and are likely to interfere with their daily activities. Among the commonly reported symptoms, we would like to highlight the high prevalence of constitutional (such as decreased exercise tolerance and fatigue) and neurological symptoms (such as headache, taste and smell disorders, cognitive impairment, memory deficits, difficulty in concentrating, vertigo, and visual impairment) (Table 4).

Several studies have highlighted the potential of anti-viral drugs, such as Nirmatrelvir/ritonavir or Remdesivir, to alleviate the development of PACS syndrome [32]. These antivirals have demonstrated efficacy in reducing the likelihood of disease progression [33] and the potential development of complications, such as fatigue, liver and cardiovascular diseases, acute kidney disease, muscle pain, neurocognitive impairment, and shortness of breath. As a result, they help prevent the impairment of work ability and quality of life. In addition, as suggested by a large study, vaccination has a protective role in the experience of PASC, effectively protecting from the risk of severe disease and reducing long-term symptoms [34].

Currently, we have no effective and recognized targeted strategies for therapy for PACS [35]. Considering the high overlap of these symptoms in relation to different conditions [36,37], a potential prospective approach could be the adoption of routine measures, such as the Work Ability Index (WAI) or the Work Productivity and Activity Impairment (WPAI), in order to provide prolonged observation on work ability and workload [38].

This study has limitations that should be considered. First, the single-center design may restrict the generalizability of the findings to similar settings. Second, the limited sample size suggests caution in the interpretation of the findings. Moreover, some patients could not be reached during the phone interviews, and the comparison of the baseline characteristics suggests that our findings may be prone to some degree of overestimation. Finally, outcome data were self-reported and collected by phone calls without performing clinical examination. Unfortunately, details about the duration of COVID-19 symptoms were not available. A future perspective would be to assess whether the duration of symptoms has an impact on the development of PASC.

However, we believe that patient-reported outcomes (such as reduced working ability and symptoms) were appropriate to investigate a patient’s perception of his/her status.

In conclusion, our findings highlight the significant impact of PASC on the working-age population, not only in terms of essential health and care aspects but also in relation to potential negative effects on work engagement and economic productivity. Our findings emphasize the importance of creating clinical pathways and guidelines for the ongoing care of patients with PASC, with a focus on working impairment.

Further research is required to identify the biomarkers and establish suitable diagnostic, prognostic, and therapeutic strategies for patients experiencing post-COVID autonomic dysfunction. Further clinical trials are needed to investigate specialized rehabilitation programs, non-pharmacologic management (such as personalized food and exercise regimes), and pharmacologic treatment (including immunotherapy). This is necessary to ensure that patients with PASC obtain the most appropriate evidence-based treatment.

Furthermore, healthcare organizations and policymakers should prioritize the implementation of structured multidisciplinary programs to reduce or eliminate the potential negative effects on work and daily life.

Addressing the long-term working impairment in COVID-19 survivors requires a multi-faceted approach that encompasses several key strategies: firstly, implementing tailored rehabilitation programs is essential. These programs should be designed specifically to address the unique challenges faced by COVID-19 survivors in their return to work. Integrating physical therapy, occupational therapy, and psychological support can enhance recovery and facilitate a smooth transition back to the workplace. Secondly, establishing long-term follow-up protocols is crucial for monitoring the health and well-being of COVID-19 survivors over an extended period. These protocols should focus on identifying and managing any lingering symptoms or functional impairments that may hinder their ability to work effectively. Thirdly, educating healthcare professionals is essential. Providing training and education to healthcare professionals can help them recognize the potential long-term working impairments in COVID-19 survivors and equip them with the knowledge and skills needed to effectively manage and support these individuals in their rehabilitation and return-to-work journey. Additionally, policy advocacy for healthcare coverage is essential. Advocating for policies that ensure adequate healthcare coverage and access to rehabilitation services for COVID-19 survivors, including coverage for specialized therapies and interventions aimed at improving their functional outcomes, is crucial in supporting their return to work and overall well-being.

## Figures and Tables

**Figure 1 viruses-16-00688-f001:**
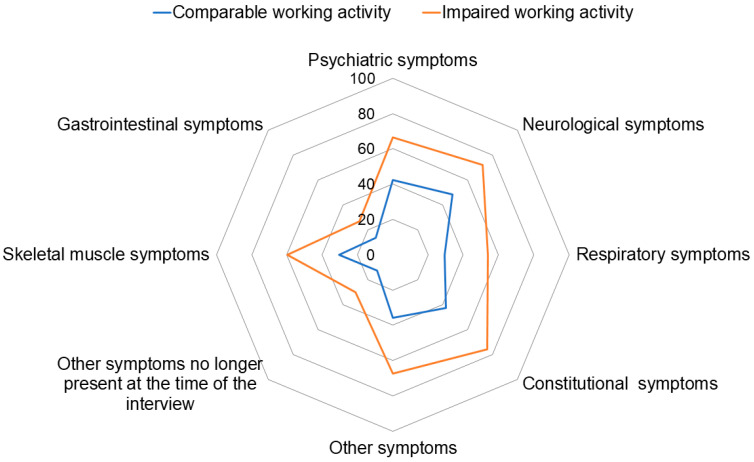
Comparison of subjects reporting comparable working ability and those reporting impaired working ability: information reported during the interview. The psychiatric symptoms included HADS depression, HADS anxiety, PTSD and sleep disorder. The neurological symptoms included headache, taste and smell disorders, cognitive impairment, memory deficits, difficulty in concentrating, vertigo and visual impairment. The respiratory symptoms included cough, dyspnea, oxygen use and chest pain. The constitutional symptoms included decreased exercise tolerance and fatigue. The skeletal muscle symptoms included myalgia and arthralgia. The gastrointestinal symptoms included abdominal pain, diarrhea, vomit and loss of appetite. The other symptoms included weight loss, pain, fever, sweats, palpitations and hair loss.

**Table 1 viruses-16-00688-t001:** Comparison of reachable and unreachable subjects.

	Reachable Subjects (*n* = 322)	Unreachable Subjects (*n* = 172)	*p*-Value
Age at hospital admission, years ^a^	46 (36.54)	49 (25–55)	0.42
Males	188 (58.4)	112 (65.1)	0.17
Hypertension	61 (18.9)	38 (22.1)	0.47
Dyslipidemia	32 (9.9)	12 (7.0)	0.34
Diabetes	28 (8.7)	15 (8.7)	0.99
Obesity (BMI > 30 km/m^2^)	58 (18.0)	24 (14.0)	0.30
Smoking habits: Current smoker Former smoker None	18 (5.6) 16 (5.0) 288 (89.4)	19/171 (11.1) 16/171 (9.4) 136/171 (79.5)	0.01
Any cancer	30 (9.3)	19 (11.0)	0.64
Any immunosuppression condition	54 (16.8)	32/170 (18.8)	0.65
Elevated D-Dimer	180/308 (58.4)	92/164 (56.1)	0.69
Elevated PCR	267/321 (83.2)	135/171 (78.9)	0.30
Elevated LDH	141/314 (44.9)	69/168 (41.1)	0.47
ICU admission	43 (13.3)	16 (9.3)	0.23
Oxygen during hospital stay	162 (50.3)	73 (42.4)	0.11
High flow nasal cannula	43 (13.3)	12 (7.0)	0.04
Non-invasive mechanical ventilation	42 (13.0)	23 (13.4)	0.99
Invasive-mechanical ventilation	12 (3.7)	8 (4.7)	0.79
Length of hospital stay, days ^a^	25 (19–40)	11 (7–17)	<0.0001

Data summarized as *n* (%) or ^a^ median (IQR).

**Table 2 viruses-16-00688-t002:** Comparison of subjects reporting comparable working ability and those reporting impaired working ability: information at admission and during hospital stay.

	Comparable Working Ability (*n* = 184)	Impaired Working Ability (*n* = 134)	*p*-Value
Age at hospital admission, years ^a^	44 (33–53)	50 (44–54)	0.001
Males	118 (64.1)	69 (51.5)	0.03
Hypertension	26 (14.1)	34 (25.4)	0.01
Dyslipidemia	13 (7.1)	19 (14.2)	0.06
Diabetes	8 (4.3)	20 (14.9)	0.002
Obesity (BMI > 30 km/m^2^)	28 (15.2)	30 (22.4)	0.13
Smoking habits: Current smoker Former smoker None	12 (6.5) 6 (3.3) 166 (90.2)	6 (4.5) 10 (7.5) 118 (88.0)	0.18
Any cancer	14 (7.6)	14 (10.4)	0.49
Any immunosuppression condition	23 (12.5)	27 (20.1)	0.09
Elevated D-Dimer	99/179 (55.3)	78/126 (61.9)	0.30
Elevated PCR	147/183 (80.3)	116 (86.6)	0.19
Elevated LDH	75/179 (41.9)	64/131 (48.9)	0.27
COVID-19 vaccination status at admission: Booster First cycle No vaccination	18/182 (9.9) 24/182 (13.2) 140/182 (76.9)	16/133 (12.0) 15/133 (11.3) 102/133 (76.7)	0.75
Corticosteroids	73 (39.7)	58 (43.2)	0.59
Remdesivir	35 (19.0)	31 (23.1)	0.45
ICU admission	26 (14.1)	15 (11.2)	0.54
Oxygen during hospital stay	83 (45.1)	77 (57.5)	0.04
High-flow nasal cannula	27 (14.7)	15 (11.2)	0.46
Non-invasive mechanical ventilation	24 (13.0)	16 (11.9)	0.90
Invasive-mechanical ventilation	7 (3.8)	5 (3.7)	0.99
Length of hospital stay, days ^a^	12 (7–18)	14 (9–23)	0.04

Data summarized as *n* (%) or ^a^ median (IQR).

**Table 3 viruses-16-00688-t003:** Multivariable analysis of predictors of impaired working ability.

	Odds Ratio (95% Confidence Interval)	*p*-Value
Age at hospital admission, years	1.02 (0.99 to 1.04)	0.05
Sex: female vs. male	1.90 (1.18 to 3.08)	0.008
Diabetes: yes vs. no	3.73 (1.57 to 9.65)	0.004
Oxygen during hospital stay: yes vs. no	1.76 (1.01 to 3.06)	0.04
Severe disease: yes vs. no ^a^	0.51 (0.26 to 1.01)	0.05

^a^ Severe disease was defined as receiving high flow nasal cannula or mechanical ventilation during the hospital stay. The initial model included a set of clinically relevant candidate predictors (age at hospital admission, sex, hypertension, diabetes, obesity, any immunosuppression condition, COVID-19 vaccination status at admission, remdesivir, oxygen during hospital stay, severe disease and length of hospital stay). The final model was selected using the AIC reduction procedure.

**Table 4 viruses-16-00688-t004:** Comparison of subjects reporting comparable working ability and those reporting impaired working ability: information reported during the interview.

Category	Comparable Working Ability (*n* = 184)	Impaired Working Ability (*n* = 134)	*p*-Value
Self-perception of the overall status ^a^	8 (7–9)	7 (6–8)	<0.0001
Psychiatric symptoms	77/182 (42.3)	89 (66.4)	<0.0001
Neurological symptoms	86/179 (48.0)	95/132 (72.0)	<0.0001
Respiratory symptoms	54 (29.3)	72/133 (54.1)	<0.0001
Constitutional symptoms	77/181 (42.5)	100/132 (75.8)	<0.0001
Skeletal muscle symptoms	56 (30.4)	80 (59.7)	<0.0001
Gastrointestinal symptoms	24/181 (13.3)	36 (26.9)	0.0004
Other symptoms	65/181 (35.9)	87/129 (67.4)	<0.0001
Other symptoms no longer present at the time of the interview	23 (12.5)	40 (29.9)	0.0002

Data summarized as *n* (%) or ^a^ median (IQR). The psychiatric category included HADS depression, HADS anxiety, PTSD and sleep disorder. The neurological category included headache, taste and smell disorders, cognitive impairment, memory deficits, difficulty in concentrating, vertigo and visual impairment. The respiratory category included cough, dyspnea, oxygen use and chest pain. The constitutional category included decreased exercise tolerance and fatigue. The skeletal muscle category included myalgia and arthralgia. The gastrointestinal category included abdominal pain, diarrhea, vomit and loss of appetite. The other category included weight loss, pain, fever, sweats, palpitations and hair loss.

## Data Availability

The datasets used and/or analyzed during the current study are available from the corresponding author at luisana.frallonardo@gmail.com on reasonable request.

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
