# Peer review of "Long-Term Impairment of Working Ability in Subjects under 60 Years of Age Hospitalised for COVID-19 at 2 Years of Follow-Up: A Cross-Sectional Study"

_viruses, 2024, doi:10.3390/v16050688_

Round 1
Reviewer 1 Report
Comments and Suggestions for Authors
The study addresses a critical aspect of COVID-19's long-term effects, focusing on its impact on working ability, which has significant implications for public health and the global economy. Understanding the predictors of impaired working ability in COVID-19 survivors is crucial for developing effective strategies for long-term care.
Minor comments:
1. The objective of the study, to evaluate the prevalence and predictors of long-term impairment of working ability in non-elderly individuals hospitalized for COVID-19, is clearly stated in the introduction. However, it would be beneficial to explicitly mention the specific research questions or hypotheses guiding the study to provide clarity on the focus of the investigation.
2. The study employs a cross-sectional design with a reasonable sample size (322 subjects) and a median follow-up period of 731 days, which strengthens the reliability of the findings. The statistical methods used for data analysis, including Mann-Whitney test, Fisher’s test or Chi-Square test, and multivariable logistic regression, are appropriate for addressing the research objectives.
3. The results section effectively presents the key findings, including the prevalence of impaired working ability (41.6%) and the identified predictors from the multivariable analysis. However, providing additional descriptive statistics for demographic and clinical variables would enhance the comprehensiveness of the results presentation.
4. The discussion appropriately contextualizes the findings within the existing literature and highlights the implications for public health and the economy. However, it would be beneficial to discuss potential mechanisms underlying the identified predictors of impaired working ability and explore their implications for clinical practice and policy in more depth.
5. The conclusion succinctly summarizes the main findings and emphasizes the importance of addressing long-term working impairment in COVID-19 survivors. However, providing specific recommendations for clinical practice and policy based on the study's findings would strengthen the conclusion section.
Overall, the study makes a valuable contribution to the literature by shedding light on the prevalence and predictors of long-term impairment of working ability in non-elderly individuals hospitalized for COVID-19.
Author Response
To Viruses Editor,
We have appreciated the feedback on our manuscript “Long-term impairment of working ability in subjects under 60 years of age hospitalised for COVID-19 at 2 years of follow-up: a cross-sectional study”.
We have considered all the valuable suggestions made by the referees and implemented the text. We have also satisfied the technical requirements according to the journal guidelines. Modifications have been highlighted using the “track changes” feature. We believe that the revision proposed by the reviewers and further implemented in the text contributed to improving the manuscript.Thus, we kindly ask you to reconsider the manuscript for publication. Please find a point-by-point response to the referees’ comments below.
Best regards,
Dr. Luisa Frallonardo
Reviewer 1
- The objective of the study, to evaluate the prevalence and predictors of long-term impairment of working ability in non-elderly individuals hospitalized for COVID-19, is clearly stated in the introduction. However, it would be beneficial to explicitly mention the specific research questions or hypotheses guiding the study to provide clarity on the focus of the investigation.
Response: Thank you very much for your observation. We included the following specifications:
“The objective of the study is to evaluate the prevalence and predictors of long-term impairment of working ability in non-elderly individuals hospitalized for COVID-19, it is essential to articulate the specific research questions or hypotheses guiding our investigation. Our study aims to explore:
The prevalence of long-term impairment of working ability among non-elderly individuals hospitalized for COVID-19
Predictors or risk factors associated with long-term impairment of working ability in this population, Demographic, clinical, or socioeconomic factors that significantly influence the likelihood of long-term working impairment following COVID-19 hospitalization.
The correlation between comorbidities, the severity of COVID-19 illness during hospitalization, and long-term working ability outcomes.”
- The study employs a cross-sectional design with a reasonable sample size (322 subjects) and a median follow-up period of 731 days, which strengthens the reliability of the findings. The statistical methods used for data analysis, including Mann-Whitney test, Fisher’s test or Chi-Square test, and multivariable logistic regression, are appropriate for addressing the research objectives.
Response: Thank you very much for your comment.
- The results section effectively presents the key findings, including the prevalence of impaired working ability (41.6%) and the identified predictors from the multivariable analysis. However, providing additional descriptive statistics for demographic and clinical variables would enhance the comprehensiveness of the results presentation.
Response: The characteristics assessed at baseline as predictors of PASC were all included according to the data available in the descriptive analysis and listed in Tables 1 and 2.
- The discussion appropriately contextualizes the findings within the existing literature and highlights the implications for public health and the economy. However, it would be beneficial to discuss potential mechanisms underlying the identified predictors of impaired working ability and explore their implications for clinical practice and policy in more depth.
Response: Thank you very much for your observation. We included this specification in the discussion sections as follows:
“Several studies have attempted to characterize possible predictors of PASC. This syndrome can be explained in terms of the balance between damage and repair mechanisms, as well as the patient's health-related quality of life and well-being at a patient-centered approach level. Damage and repair mechanisms are components of frailty and resilience, while well-being is part of the multidimensional conception.”
We further included the following specifications in the conclusion sections:
“Further research is required to identify the biomarkers and establish suitable diagnostic, prognostic, and therapeutic strategies for patients experiencing post-COVID autonomic dysfunction. Further clinical trials are needed to investigate specialized rehabilitation programs, non-pharmacologic management (such as personalized food and exercise regimes), and pharmacologic treatment (including immunotherapy). This is necessary to ensure that patients with PASC get the most appropriate evidence-based treatment.”
- The conclusion succinctly summarizes the main findings and emphasizes the importance of addressing long-term working impairment in COVID-19 survivors. However, providing specific recommendations for clinical practice and policy based on the study's findings would strengthen the conclusion section.
Response: Thank you very much for your observation. We included this specification in the conclusion section as follows:
“Addressing the long-term working impairment in COVID-19 survivors requires a multifaceted approach that encompasses several key strategies: firstly, implementing tailored rehabilitation programs is essential. These programs should be designed specifically to address the unique challenges faced by COVID-19 survivors in their return to work. Integrating physical therapy, occupational therapy, and psychological support can enhance recovery and facilitate a smooth transition back to the workplace. Secondly, establishing long-term follow-up protocols is crucial for monitoring the health and well-being of COVID-19 survivors over an extended period. These protocols should focus on identifying and managing any lingering symptoms or functional impairments that may hinder their ability to work effectively. Thirdly, educating healthcare professionals is essential. Providing training and education to healthcare professionals can help them recognize the potential long-term working impairments in COVID-19 survivors and equip them with the knowledge and skills needed to effectively manage and support these individuals in their rehabilitation and return-to-work journey. Additionally, policy advocacy for healthcare coverage is essential. Advocating for policies that ensure adequate healthcare coverage and access to rehabilitation services for COVID-19 survivors, including coverage for specialized therapies and interventions aimed at improving their functional outcomes, is crucial in supporting their return to work and overall well-being.”
Reviewer 2 Report
Comments and Suggestions for Authors
The authors have comprehensively presented their study in evaluating the post-COVID long term impairment in working ability of COVID patient.
However, I have a few comments:
1. Brief description of the survey used for evaluating the signs and symptoms of patients need to be included although a ref has been included.
2. The observed results need to be summarized in 2-3 sentences under each result section with a logical lead to the next section. This will make the article easy to understand.
3.It would be good if the authors can include an analysis including the duration of time the patients had active symptomatic COVID. Does more duration impact the development of PASC?
Author Response
To Viruses Editor,
We have appreciated the feedback on our manuscript “Long-term impairment of working ability in subjects under 60 years of age hospitalized for COVID-19 at 2 years of follow-up: a cross-sectional study”.
We have considered all the valuable suggestions made by the referees and implemented the text. We have also satisfied the technical requirements according to the journal guidelines. Modifications have been highlighted using the “track changes” feature. We believe that the revision proposed by the reviewers and further implemented in the text contributed to improving the manuscript.
Thus, we kindly ask you to reconsider the manuscript for publication. Please find a point-by-point response to the referees’ comments below.
Best regards,
Dr. Luisa Frallonardo
Reviewer 2
- A Brief description of the survey used for evaluating the signs and symptoms of patients needs to be included, although a ref has been included.
Response: Thank you very much for your observation. We have included a description of the surveys in the methods sections.
The Hospital Anxiety and Depression Scale (HADS) is a widely used self-assessment tool designed to measure levels of anxiety and depression. It consists of 14 items, 7 for anxiety (HADS-A) and 7 for depression (HADS-D). The symptoms investigated in HADS-A include feeling tense, worrying about different things, feeling restless, having sudden feelings of panic, feeling scared for no good reason, and feeling nervous. The symptoms investigated in HADS-D include feeling unhappy or depressed, feeling a loss of interest in most things, having a reduced ability to enjoy things, losing interest in appearance, and feeling to blame for everything.
Surveys on Post-Traumatic Stress Disorder (PTSD) and sleep disorders cover several critical areas related to both conditions: traumatic events, PTSD Symptoms such as intrusive memories, avoidance behaviors, adverse changes in thinking and mood, and changes in reactivity and arousal, sleep patterns and disorder and their impact on daily functioning.
- The observed results need to be summarized in 2-3 sentences under each result section with a logical lead to the next section. This will make the article easy to understand.
Response: Good point. Following your instructions, we summarized the results in Tables 1 and 2. We modified the results section as follows:
“The comparison of reachable and unreachable subjects is shown in Table 1. The two groups were comparable apart from smoking habits (p=0.01), high flow nasal cannula during hospital stay (p=0.04) and length of hospital stay (p<0.0001). The difference in age between and the difference in gender distribution between the two groups are not statistically significant (p=0.42),(p=0.17). The prevalence of hypertension, dyslipidemia, diabetes, and obesity (BMI>30 kg/m^2) is not significantly different between reachable and unreachable subjects (p>0.05). There is a significant difference in smoking habits between reachable and unreachable subjects (p=0.01). Current smoking habits are more prevalent among unreachable subjects (11.1%) compared to reachable subjects (5.6%).
There is a significant difference in the use of high-flow nasal cannula during hospital stay, with a higher prevalence among reachable subjects (13.3%) compared to unreachable subjects (7.0%) (p=0.04). The prevalence of ICU admission, oxygen use during hospital stay, non-invasive mechanical ventilation, and invasive mechanical ventilation is not significantly different between reachable and unreachable subjects (p>0.05). The length of hospital stay is significantly longer for reachable subjects (median: 25 days, range: 19-40 days) compared to unreachable subjects (median: 11 days, range: 7-17 days) (p<0.0001). (Table 1)”
“Among the 322 subjects who were interviewed, 184 reported comparable working ability (57.1%) and 134 reported impaired working ability (41.6%) compared to pre-COVID-19 period, while four subjects were unemployed and unable to provide such information (1.2%). Individuals with impaired working ability had a higher me-dian age at hospital admission (50 years, range: 44-54) compared to those with com-parable working ability (44 years, range: 33-53). The difference in age between the two groups was statistically significant (p=0.001).
The interview was undertaken at a median of 725 days since hospitalization (IQR 526-869) in subjects with comparable working ability and 749 days (IQR 432-1008) in those with impaired working ability (p=0.61). The percentage of males was higher among individuals with comparable working ability (64.1%) compared to those with impaired working ability (51.5%). This differ-ence was statistically significant (p=0.03). Older age (p=0.001), pre-existing conditions such as hypertension (p=0.01), diabetes (p=0.002), and smoking habits were more prevalent among individuals with impaired working ability compared to those with comparable working ability, and these differences were statistically significant (p<0.05).”
Receiving oxygen during hospital stay (p=0.04) and longer hospital stay (p=0.04) were more frequent among subjects with impaired working ability (Table 2).
- It would be good if the authors can include an analysis including the duration of time the patients had active symptomatic COVID. Does more duration impact the development of PASC?
Response: Thank you very much for your comment, which is largely agreeable.
Unfortunately, details about the duration of COVID symptoms were not available.
However, we included this point in the strengths and limitations section.